# A Dual-Band 47-dB Dynamic Range 0.5-dB/Step DPA with Dual-Path Power-Combining Structure for NB-IoT

**DOI:** 10.3390/s22093493

**Published:** 2022-05-04

**Authors:** Reza E. Rad, Sungjin Kim, Younggun Pu, Yeongjae Jung, Hyungki Huh, Joonmo Yoo, Seokkee Kim, Kang-Yoon Lee

**Affiliations:** 1Department of Electrical and Computer Engineering, Sungkyunkwan University, Suwon 16419, Korea; reza@skku.edu (R.E.R.); sun107ksj@skku.edu (S.K.); hara1015@skku.edu (Y.P.); yjjung@skaichips.co.kr (Y.J.); gray@skaichips.co.kr (H.H.); jmyoo@skaichips.co.kr (J.Y.); skkim@skaichips.co.kr (S.K.); 2SKAIChips Co., Ltd., Suwon 16419, Korea

**Keywords:** DPA, NBIoT, binary-weighted PA, dynamic range, fine-tuning

## Abstract

This paper presents a digital power amplifier (DPA) with a 43-dB dynamic range and 0.5-dB/step gain steps for a narrow-band Internet of Things (NBIoT) transceiver application. The proposed DPA is implemented in a dual-band architecture for both the low band and high band of the frequency coverage in an NBIoT application. The proposed DPA is implemented in two individual paths, power amplification, and power attenuation, to provide a wide range when both paths are implemented. To perform the fine control over the gain steps, ten fully differential cascode power amplifier cores, in parallel with a binary sizing, are used to amplify power and enable signals and provide fine gain steps. For the attenuation path, ten steps of attenuated signal level are provided which are controlled with ten power cores, similar to the power amplification path in parallel but with a fixed, small size for the cores. The proposed implementation is finalized with output custom-made baluns at the output. The technique of using parallel controlled cores provides a fine power adjustability by using a small area on the die where the NBIoT is fabricated in a 65-nm CMOS technology. Experimental results show a dynamic range of 47 dB with 0.5-dB fine steps are also available.

## 1. Introduction

Recently, by spreading several applications for Internet of Things (IoT) devices, many studies have been done for this useful communication application. The major waves in IoT are caused by fifth-generation (5G) mobile communications systems. Innovations like automated lifecycle management, software-defined networking, network slicing, and cloud-optimized distributed network applications are expected of 5G and IoT. Narrow-band IoT (NB-IoT), similar to LTE Cat-M, is often viewed as the second generation of LTE chips built for IoT applications. It completes the cost and power-consumption reduction. However, NB-IoT uses DSSS modulation instead of LTE radios.

Nonetheless, NB-IoT is considered as an option with a potentially lower cost due to the elimination of the need for a gateway. Whereas other infrastructures usually have gateways that aggregate sensor data, which then communicate with the primary server, the data of the NB-IoT sensor is sent directly to the primary server. This is the reason for the massive investing by large manufacturers in NB-IoT. Due to this importance, enhancing the hardware of NB-IoT is very critical, and several studies are done to aid its requirements. NBIoT, compared to human-oriented 4G technologies, has key design benefits in terms of increased coverage, enhancement of power saving, and a reduced set of functionalities. Therefore, a longer battery life, allowing the connectivity in challenging positions for devices, and reducing device complexity [1] are benefits.

One the critical parts of any RF transceiver is the transmitter. Due to the coverage of B1-B5, B8, B11-B14, B17-B20, B25-B26, B28, B31, B66, B70-B74, and B85 in NB-IoT, gain steps are required to be implemented precisely. The up-link (UL) frequency ranges of 0.699–0.915 GHz and 1.71–1.98 GHz are considered as the low- and high-frequency bands, respectively. One of the aspects of the DPA that must be concerned is the gain-controllability of the PA. Providing a dynamic range of output power with fine gain steps is demanding.

In [2], a DPA is proposed which only covers the low bands from 0.75 GHz to 0.96 GHz by using a reconfigurable structure providing 3-dB gain steps, but it does not cover the high bands. In [3], a multi-tapped balun is proposed to provide the coarse gain steps while the fine gain steps are implemented by using the tuning capacitors in parallel with the balun. The drawback of this work is the limited number of the gain steps and very low dynamic range even though it covers both the low- and high-frequency bands. A two-stage differential cascode structure is proposed in [4], which can adjust the output power by using the bias control. It is known that providing an output power range by using bias control does not satisfy the NB-IoT’s demanding gain control requirement. In [5], a novel structure is proposed by using a class-G efficiency enhancement with a high-current power supply requirement and SC cells. This work proposed two on-chip baluns. Despite the mentioned parameters, the area occupation (1.3 mm × 1.4 mm) is larger than the dictated area for our design, which has 1.2 mm × 0.9 mm available area for the design. In [6], a dual-band transmitter is proposed in two transmit paths for low and high bands. A two-stage differential and cascade structured cores are used in parallel with each other that are enabled to provide the gain steps. Therefore, the number of the available gain steps are obtained by multiplying the number of the driver cores and the power cores. Due to the parasitic issues of parallel components in the high frequencies, the number of the parallel cores will be limited, which might result in an inconsistent gain step, and providing linear gain steps in a wide dynamic range would be difficult.

In this paper, a dual-path structure is proposed for the DPA. This structure will provide a very finely tuned adjustability as small as 0.5 dB, and the required gain steps are considered as 1 dB/step, which is smaller than the 3 dB/step in the other works [2]. In comparison with similar works [2,3,4], the gain steps are provided with a different technique with very finely tuned results. Therefore, by having finer gain steps and tuning (0.5 dB/step), a consistent power level for different bands is obtained, and a power mismatch between the bands decreases. The gain adjustability of the proposed DPA provides the required flexibility to reconfigure and operate at the f_c_ of whole the B1–B5, B8, B11–B14, B17–B20, B25–B26, B28, B31, B66, B70–B74, and B85 bands in a NB-IoT transceiver. A dual-band structure is proposed for the DPA to cover all the low and high bands individually. Moreover, to provide the required high-dynamic output power range, a dual-path structure having both the power amplification and power attenuation is implemented.

## 2. The Proposed Dual-Band DPA Structure

Figure 1 shows the top block diagram of the proposed DPA. The DPA is implemented in two individual paths for low band (LB) and high band (HB). The input RF signal for both the bands are coming from a conventional active up-mixer. For each power path, a DPA is designed with a custom-made balun. Every DPA is controlled by 20 control bits to provide the fine gain-steps. At a glance, every DPA is formed by 20 differential cascode sub-PA cores, which is discussed in the next section. The digital controls, turn on/off the power cores by turning on/off the cascode stage’s bias (V_bc<19:0>_). Even though the operation of the DPA is similar for both the bands, an optimization is required over the circuits and the baluns in respect to the desired frequency band.

### 2.1. The Proposed Dual-Path DPA Implementation

The structure for the DPA is shown in Figure 2. The structure is proposed in a two- path implementation for the high-gain and low-gain paths. Because the structure must produce a power level in the range of −40 dBm to 23 dBm for a fixed value of input power (P_DPA_IN_), both the attenuation and amplification processes are required. If P_ADPA_IN_ < P_OUT_ (the level of the desired output power based on the control word), the high-gain (HG) or amplification path is on and the low-gain (LG) or attenuation path is off. A 20-digit control word is used to adjust the output power, whereas 10 MSBs are related to the HG path and 10 LSBs are used to control the LG path. As is shown in Figure 2, the control bits in the control center control the cascode bias voltage of all the power cores. When a cascode bias voltage is 0, the current of that core is 0 and it is not operating. Therefore, the reconfiguration mechanism of the DPA is an on/off operation for both the LG and HG paths.

### 2.2. The Attenuation Path

When an output power level (P_OUT_) less than P_DPA_IN_ is required, the RF signal is fed to the low-power path, and the high-power path is off. Based on the digital control word, the level of the attenuation is controlled by the attenuation unit. For the smaller control words, the output power must be smaller; therefore, the level of the attenuation must be higher as well. The low gain path is by the attenuation unit and ten differential cascode cores. Every low-gain core (LG_core) is controlled by its cascode voltage (VCAS_LG) and 10 LSB digits of the control word.

The attenuation path circuit is shown in Figure 3, including the attenuation unit. It is shown that by 10 control digits, 1024 connection combinations are available. The key challenging point for such an implementation is how to combine several different values of signal level together with a shared output node. To combine the various power levels at the shared output node, ten differential cascode cores are implemented with a same and small sizing in comparison with the high-gain path’s transistors. Because the input stage is formed by a common-source amplifier, the drain current of MLG-1<1:10> and MLG-2<1:10> transistors are proportional to the applied attenuated signal level at the gate source of every transistor. Therefore, the power combination process is performed by the drain currents of every core at the output node as below:
*i_OUT_* = *i_D1_* + *i_D2_* + … + *i_D10_*
(1)
 where *i_D_*_1_*–i_D_*_10_ are the output drain RF signal current of one side the differential cores. Therefore, based on the control word, a combination of ten signals, RF_IN_ATT<1:10>, are transformed to the output current, which is transferred to the secondary side of the balun toward the antenna.

### 2.3. The Amplification Path

Figure 4 illustrated the schematic of the completed proposed DPA including the LG and HG paths. It is shown that the amplification path is formed with 10 differential cascode power amplifier cores, which are binary sized from HG-Core #1 to HG-Core #10. Therefore, the MSBs are going to the larger cores. The power-combining technique is like the attenuation path and is performed by the shared output node at the output. Both the LG and HG paths are connected to a shared custom-made transformer. This structure is similarly used for both the LB and HB bands.

### 2.4. Design of the Output Baluns and Top Layout

Figure 5 shows the top layout of the proposed dual-band DPA. It occupies a 0.6-mm by 1.2-mm die area, totally. Figure 6 and Figure 7 show the designed center tapped baluns with their quality factor (Q) and inductance (L) for the HB and LB bands, respectively. The center-tap (CT) is used to bias the drain of the cascode transistors (MHG-3<1:10>, MHG-4<1:10>, MLG-3<1:10> and MLG-4<1:10>). As it is shown in the top layout, output capacitors (C_OUT_) are added in the primary sides of the baluns to place the self-resonance frequency (SRF) of the baluns at the center of each band. Figure 8a,b shows the optimized AC response of the DPAs after adding the C_OUT_ at the primary side of the baluns for the low and high bands, respectively. Due to the compactness of the area, the core of the DPAs are designed with a high compactness in terms of the layout.

### 2.5. Gain-Steps and Dynamic Range

Simulation results shows a possible 2^20^ possibilities for the output power levels. By selecting the control words between these possibilities, very accurate fine-gain steps are feasible. This feasibility is tested and proofed in both the simulation and measurement results. Figure 9 shows the simulated gain steps of the DPA by using a selected arrangement of the control codes. It is shown that the proposed DPA provides a 47-dB dynamic range with 0.5 dB/step fine-gain steps. Figure 10 shows the post-simulation results for IIP3 and OIP3 of the DPA. An input-referred 1-dB comparison point (P1dB_IN_) and output-referred 1-dB comparison point (P1dB_OUT_) are shown in Figure 11.

Due to the nonlinear relationship between the current and voltage, using 20 control bits, $2^{20}$ power levels are possible (1,048,576 states) that are not in a linear order by themselves. The huge number of the possibilities in a limited dynamic range of power levels, provides very close power values which are used for fine tuning. In addition, a precise measurement must be done to obtain the corresponding gain control codes for the linear gain steps in the full range of output power range.

## 3. Experimental Results

The proposed DPA is designed and implemented in a 65-nm RF CMOS technology. Figure 12 shows the chip micrograph of the fabricate NB-IoT IC, which HB and LB DPAs are known, similar to the top layout in Figure 5. The measurement setup of the test board and the device under test (DUT) is shown in Figure 13. The DPA is designed to provide the dynamic range and gain steps for a fixed external PA. Figure 14 shows the test board and the designed DPAs is driving an external PA. Measurement results show the fine-gain steps as small as 0.5 dB for the gain steps. Due to the requirements, 1 dB/step is chosen for the measurement whereas the fine adjustment is performed to provide the accurate gain steps. Even though the dynamic range of the DPA itself achieved 47 dB similar to the simulations, due to the power level higher than 23 dBm, 43 dB of this dynamic range is used. The input of the DPA is provided by the transmitter path through the DAC, BBA, and the up-conversion mixer.

The measured gain steps and the used dynamic range of the LB and HB bands are shown in Figure 15 and Figure 16, respectively. It is shown that the min/max current consumption of the DPAs are 4 mA/36 mA and 3.43 mA/25 mA. The consistency between the bands is because of the fine-gain tuning of the DPA as low as 0.5 dB. Therefore, due to this novelty, providing very accurate consistence gain steps is necessary. Figure 17a,b show the maximum measured power at the antenna port for LB and HB bands, respectively, whereas a maximum value of 27 dBm is also feasible, but due to the over spec condition 4 dB of the dynamic range is not intentionally used.

Table 1 shows a comparison between this work and similar works in terms of the DPA performance. One of the design limitations for this work is the restricted low voltage of the power supply, 1 V, which is lower in comparison with the other works. Also, the proposed work covers the whole of the NB-IoT bands in comparison with [2] which only covers the lower frequency bands of NB-IoT. This frequency coverage in parallel with the compact design of the custom-made integrated baluns, and the baluns’ optimization by the output capacitors, provides a fully integrated implementation. The other contribution of this work compared to the other works is the precise fine gain steps as small as 0.5 dB for power adjustments. This feature provides consistency among the frequency bands when having the same power levels with the same gain steps are desired, as was shown in Figure 13 and Figure 14. Finally, the proposed DPA shows a higher linearity in terms of in-band IIP3. The following figure of merit (F.O.M.) is defined to compare the mismatch between the power level in the different bands as follows:(2)$$F.O.M.=\left(1-\left|\frac{P_{OUT\_BAND2}}{P_{OUT\_BAND1}}\right|\right)\times 100\%$$
where $P_{OUT\_BAND1}$ must be the larger value to satisfy $P_{OUT\_BAND1}>P_{OUT\_BAND2}$. This F.O.M. reflects the role of the fine gain steps to provide the consistency between the bands. For this case, because the previous works have not provided any data comparing the power consistency among the bands for the proposed dual-band implementation, the comparison is performed between the low and high bands, resulting in a maximum 1% mismatch of power level which is better in comparison with [5,8,9].

## 4. Conclusions

In this paper, a novel dual-band binary-weighted DPA is designed and implemented for an NB-IoT transceiver. The proposed structure for the DPA features the NB-IoT transmitter with fine gain steps. The DPA is fabricated in a 65-nm CMOS process, and experimental results show the 0.5-dB/step gain steps with 47-dB dynamic range. Due to the required specs, 43 dB of this dynamic range is used. Power consistency among the low and high bands, and their sub-bands are performed that show a 1% power mismatch between the bands, which is the best result in the literature.

## Figures and Tables

**Figure 1 sensors-22-03493-f001:**
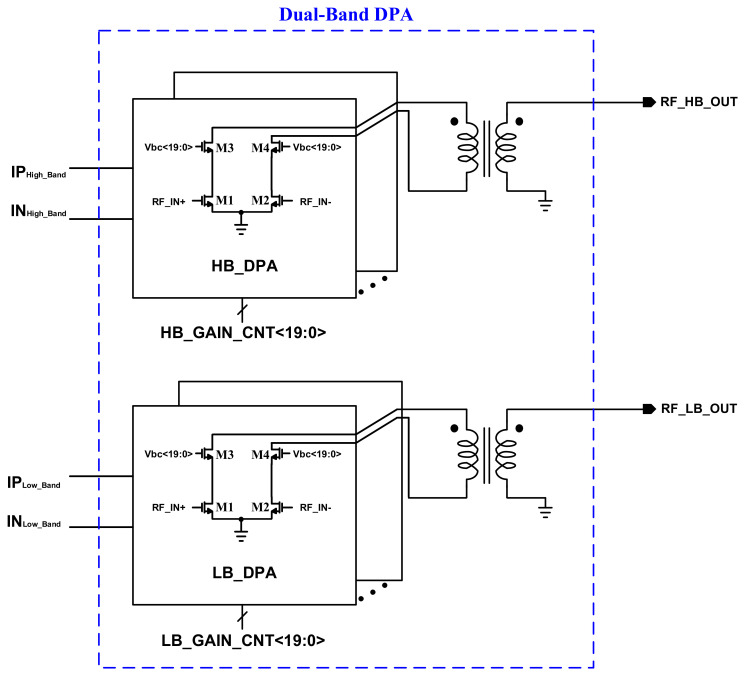
The top block-diagram of the proposed dual-band structure of the DPA covering whole the sub-bands of the NB-IoT application.

**Figure 2 sensors-22-03493-f002:**
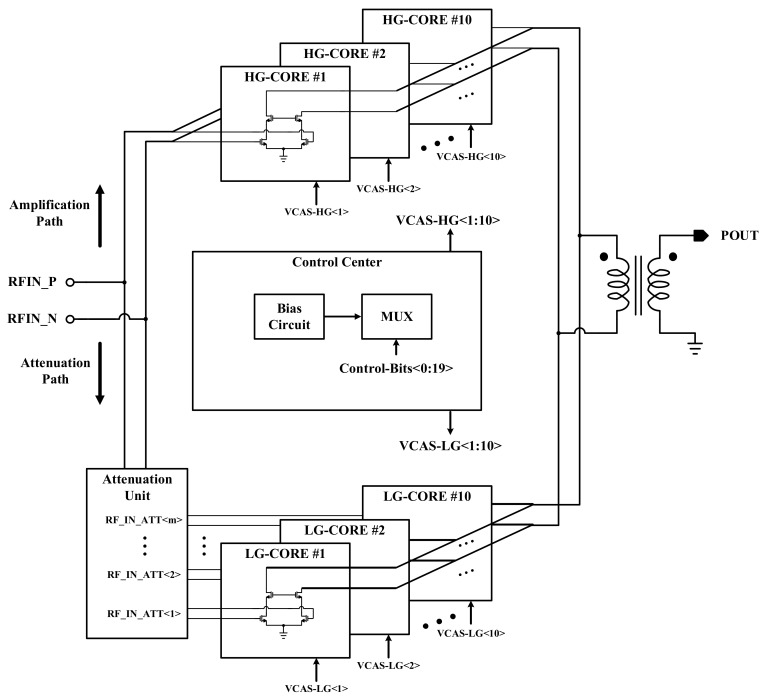
The proposed power combining DPA structure in a dual path of power amplification and attenuation implementation.

**Figure 3 sensors-22-03493-f003:**
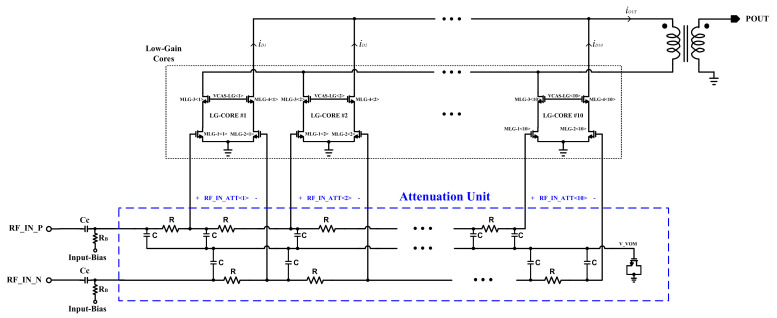
The proposed attenuation and power combining path of the DPA.

**Figure 4 sensors-22-03493-f004:**
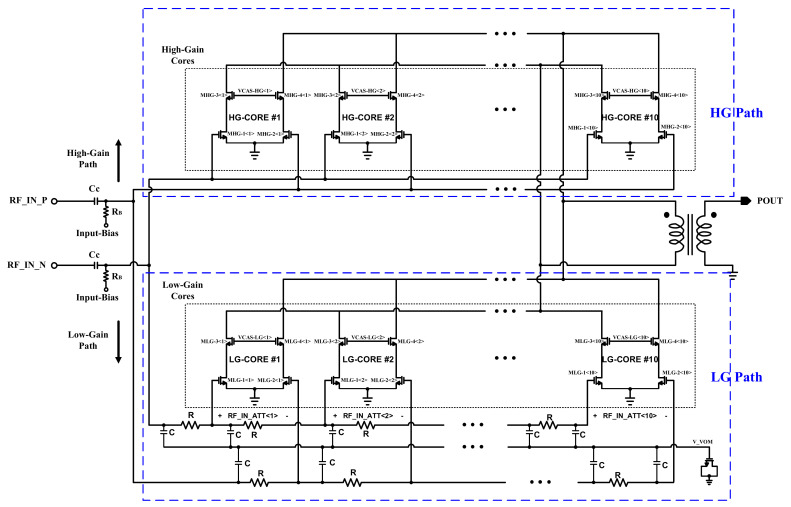
The top schematic of the proposed power combining DPA structure in a dual path of power amplification and attenuation implementation.

**Figure 5 sensors-22-03493-f005:**
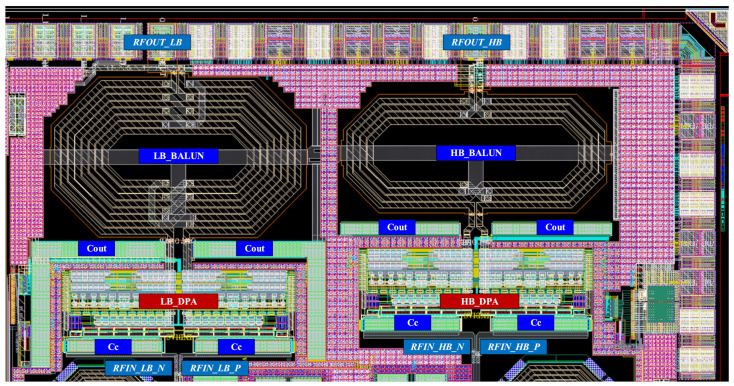
The top layout of the dual-band DPA.

**Figure 6 sensors-22-03493-f006:**
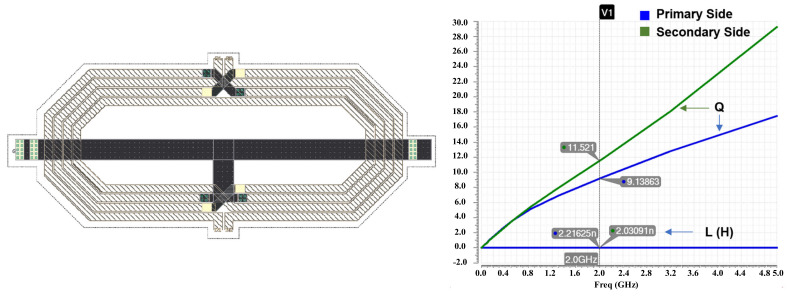
The designed center-tapped (CT) HB balun with Q and L simulations for both the primary and secondary sides.

**Figure 7 sensors-22-03493-f007:**
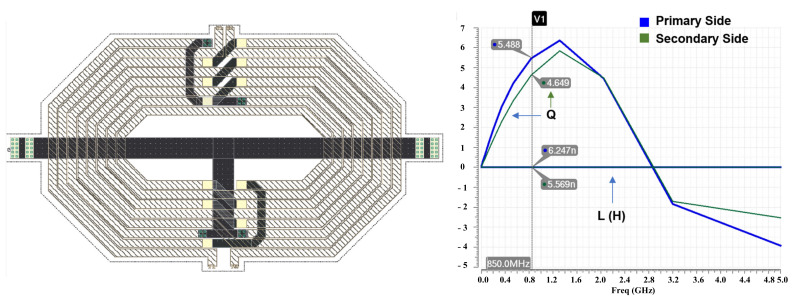
The designed center-tapped LB balun with Q and L simulations for both the primary and secondary sides.

**Figure 8 sensors-22-03493-f008:**
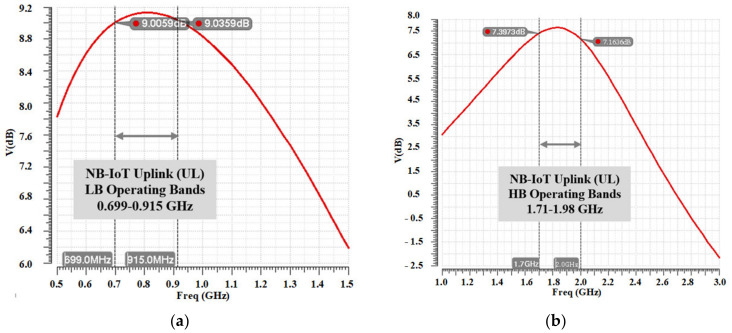
Optimized AC response for (**a**) low and (**b**) high bands.

**Figure 9 sensors-22-03493-f009:**
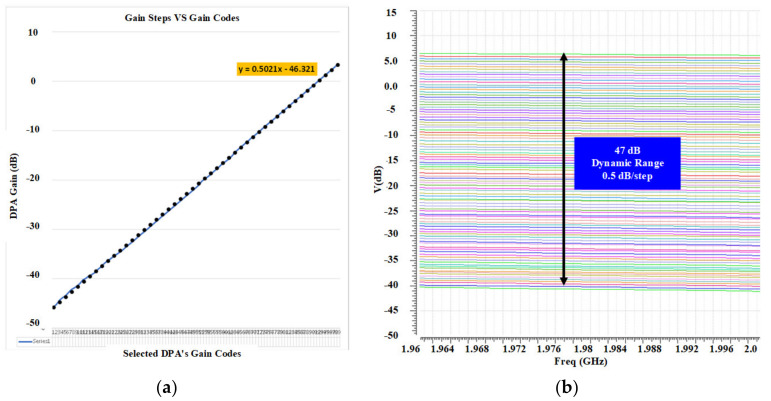
Post-layout simulation results illustrating (**a**) the linear gain steps by the selected gain control codes and (**b**) for the gain-steps accuracy and the dynamic range.

**Figure 10 sensors-22-03493-f010:**
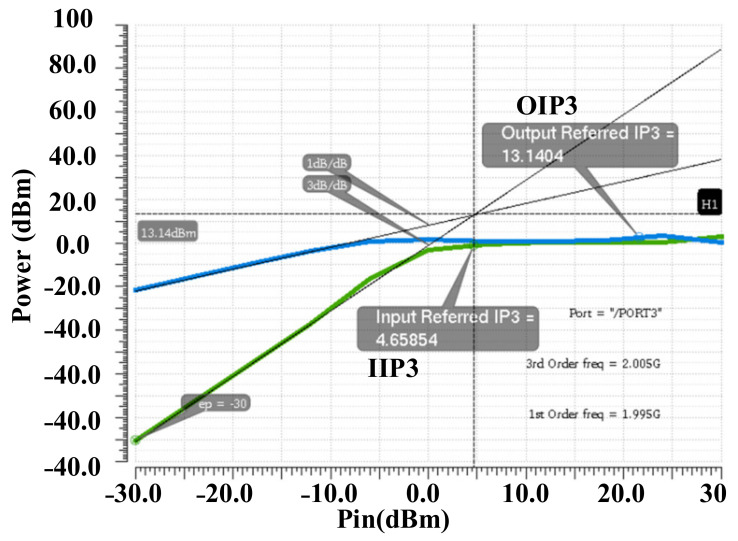
Post-layout simulation results for IIP3 and OIP3 of the DPA.

**Figure 11 sensors-22-03493-f011:**
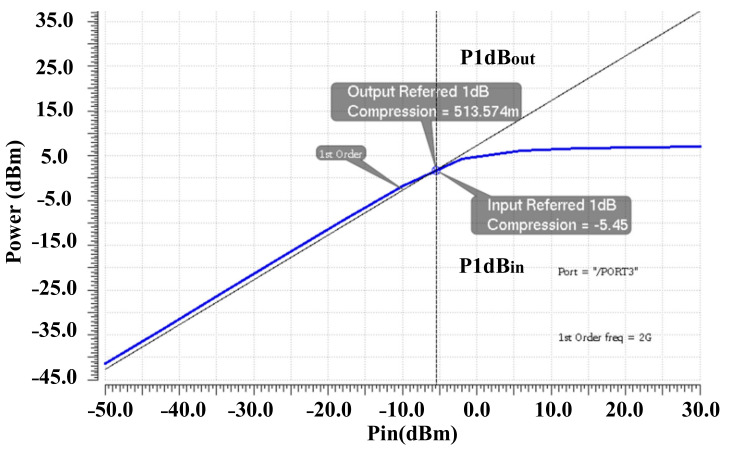
Input-referred 1-dB comparison point, P1dB_IN_, and output-referred 1-dB comparison point, P1dB_OUT_.

**Figure 12 sensors-22-03493-f012:**
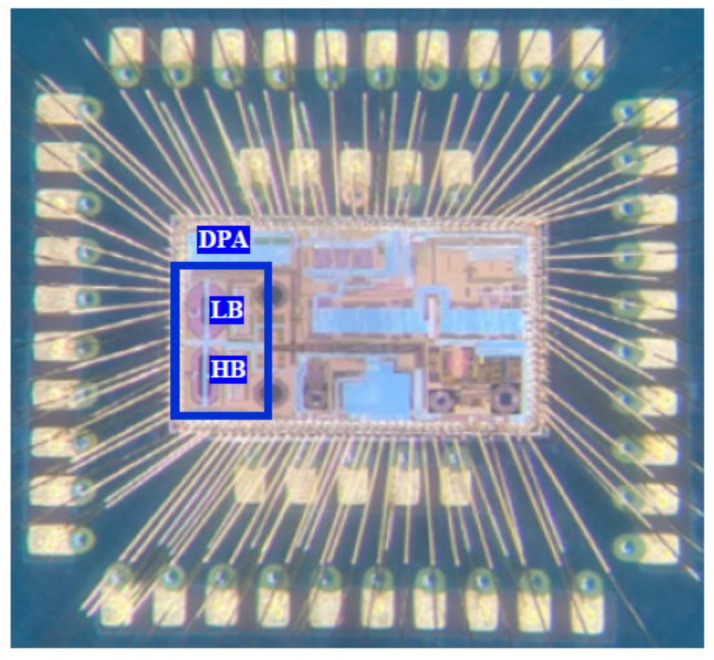
Chip-micrograph of the fabricated IC.

**Figure 13 sensors-22-03493-f013:**
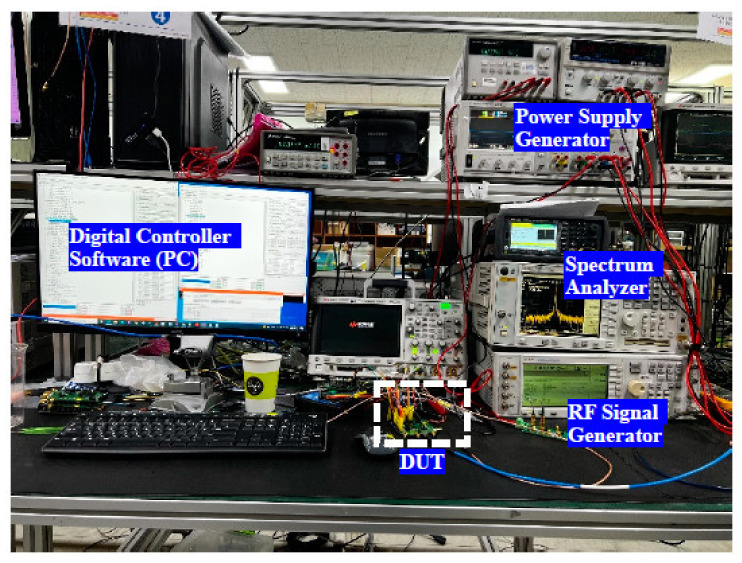
Measurement setup for the DPA (DUT) test.

**Figure 14 sensors-22-03493-f014:**
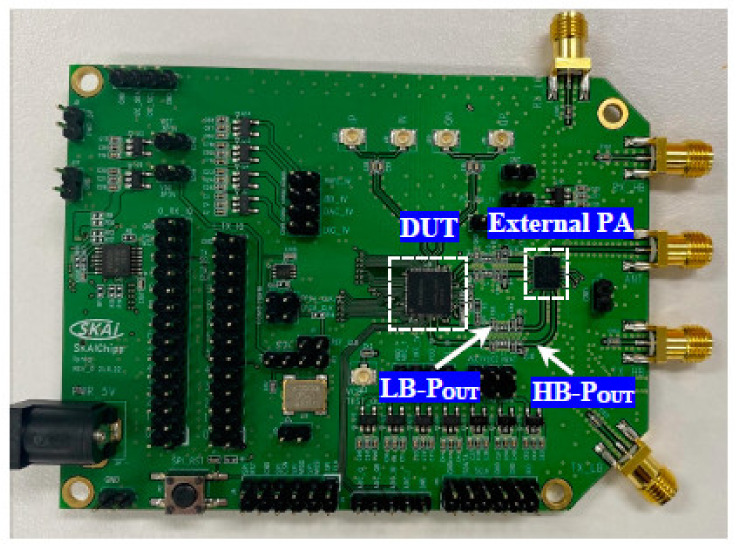
The test board of the DPA (DUT) showing the LB and HB paths and the external PA.

**Figure 15 sensors-22-03493-f015:**
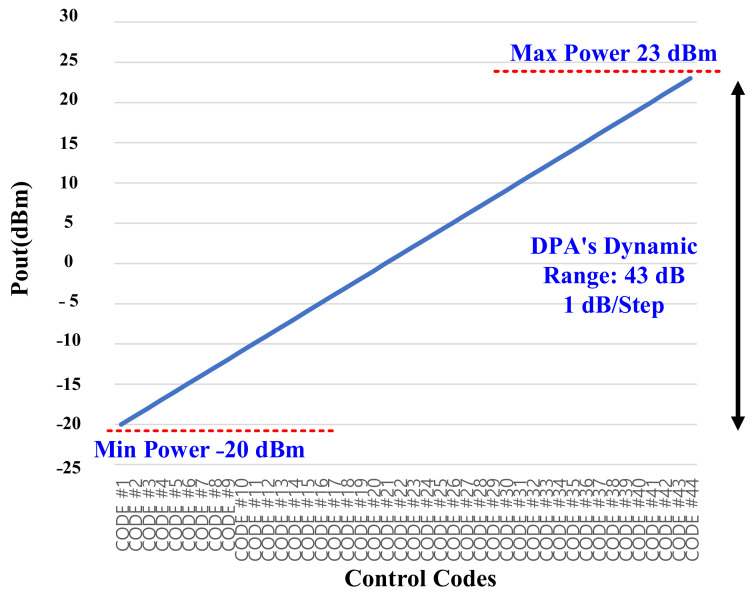
The measured gain steps and dynamic range for the low band (the results are standardized for the application using the 0.5 dB fine-gain adjustability and reducing the maximum power from 27 dBm to 23 dBm).

**Figure 16 sensors-22-03493-f016:**
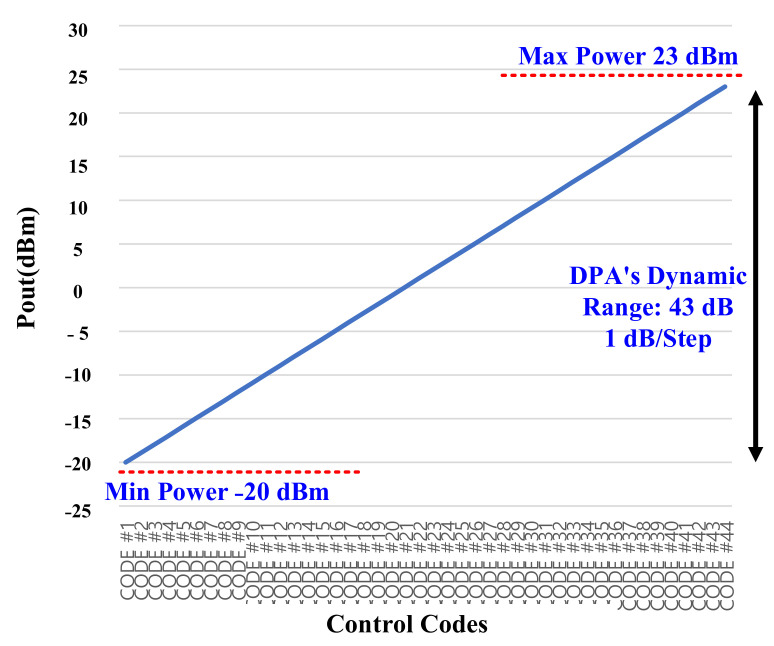
The measured gain steps and dynamic range for the high band (the results are standardized for the application using the 0.5 dB fine-gain adjustability and reducing the maximum power from 27 dBm to 23 dBm).

**Figure 17 sensors-22-03493-f017:**
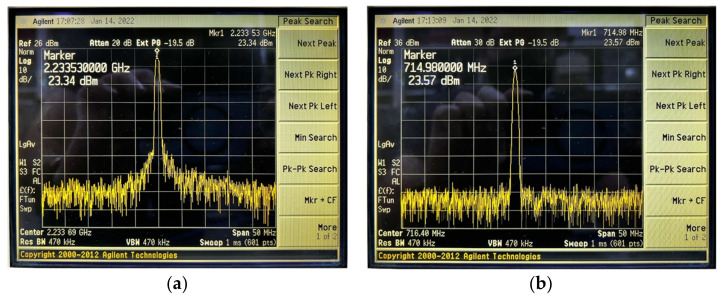
The maximum 23 dBm consistent power level for the (**a**) low and (**b**) high bands.

**Table 1 sensors-22-03493-t001:** Comparison table over various TIA structures.

Parameters	[2]	[5]	[7]	[8]	[9]	This Work
Process Technology	CMOS 0.18 um	CMOS 40 nm	CMOS 0.18 um	InGaP/GaAs	CMOS 0.13 um	CMOS 65 nm
Supply Voltage (V)	2	2.2	1.55	3.3	NA	1
Frequency Range (GHz)	0.75–0.96	0.699–0.915	0.36–0.54 2.36–2.5	0.8 1.9 2.4	0.76–0.96 1.3–1.6 1.7–2.34 2.35–2.81	0.699–0.915 1.71–1.98
Die Area (mm^2^)	NA	1.8	NA	NA	4.5	1.08
PAE_MAX_ (%)	44.5	33.3	40	35	5.5	34.7
PA Core Configuration	Inverse Class-D	Class-G	Class-AB/B/C	Class-E	NA	Binary-Weighted Class-AB
Application Bands	NB-IoT Low Bands	LTE NB-IoT and eMTC	IEEE 802.15.6	LTE	LTE	Dual Band Low Bands And High Bands
Integration	External Balun	Fully Integrated (In-Chip Baluns)	External Balun	No External Components	NA	Fully Integrated (In-Chip Baluns)
Gain-Steps	3 dB/Step	NA	NA	NA	Tunable	0.5 dB/Step
F.O.M.	NA	9.2%	NA	5.4%/10.4%	19%/4.7%	1%

## Data Availability

Not applicable.

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
