# Peer review of "A Dual-Band 47-dB Dynamic Range 0.5-dB/Step DPA with Dual-Path Power-Combining Structure for NB-IoT"

_sensors, 2022, doi:10.3390/s22093493_

Round 1
Reviewer 1 Report
The structural parts of this scientific article are:
- Introduction is very short (literature sources 1-4): In this paper a dual-path structure is proposed for the DPA which is providing a very fine tune adjustability as small as 0.5 dB while the required gain steps are considered 1 dB/step which is smaller than 3 dB/step in the other works. But it is not analyzed why other works did not follow this, where is the general technical problem?
- The Proposed Dual-Band DPA Structure
This is the main section for modeling and computation DPA (Fig.1-Fig.10).
- Experimental Results
This section describes the measurement results and photos of parts of the project (Table 1; Fig.11-Fig.15). The proposed DPA is designed and implemented in a 65-nm RF CMOS technology. Figure 11 shows the chip micrograph of the fabricate NB-IoT IC, which HB and LB DPAs are known similarly to the top-layout in Figure 5 (Table 1; Fig.11-Fig.15). Table 1 shows a comparison between this work and similar works in terms of the DPA performance. The article contains in Table 1 the mistake: “Process technology” - 0.65 um (need - 65 nm).
- Conclusion
In this part briefly describe the main results, which are also in the title of the article. Basically, this is not a conclusion, but a statement of key outcomes.
References [1-7].
Interesting work, the article contains all the necessary parts from the theoretical proposal to the realization and practical measurements.
Suggestions:
1. The title of the article is too long: too much details.
2. The authors present only a very small number of articles in the introduction to prove the relevance of the problem they have raised.
3. Thus, TABLE 1's comparisons with the authors' work are inaccurate. It needs to be expanded, after all, multiband TIAs are plentiful, and with 65nm CMOS technology. The article contains in Table 1 the mistake: “Process technology” - 0.65 um (need - 65 nm).
4. The authors do not describe what technological errors affect the possible errors in such DPAs.
Author Response
Concern # 1: The title of the article is too long: too much details.
Author response: The reviewer’s comment is concerned and the title is simplified in comparison with the first submission.
Author action: We updated the manuscript by shorting the title and removing additional details to “A Dual-Band 47 dB Dynamic Range 0.5 dB/Step DPA with Dual-Path Power-Combining Structure for NB-IoT”.
Concern # 2: The authors present only a very small number of articles in the introduction to prove the relevance of the problem they have raised.
Author response: The reviewer’s concern is followed and the introduction is revised.
Author action: We updated the manuscript by adding more references (references 5 and 6), analysis and descriptions for the prior works on page 2, the second and third paragraphs, and the 9th and 5th lines, respectively.
Concern # 3: Thus, TABLE 1's comparisons with the authors' work are inaccurate. It needs to be expanded, after all, multiband TIAs are plentiful, and with 65nm CMOS technology. The article contains in Table 1 the mistake: “Process technology” - 0.65 um (need - 65 nm)
Author response: Following the comment, Table 1 is revised.
Author action: We updated the manuscript by correcting the 0.65 um to 65 nm in the comparison table.
Concern # 4: The authors do not describe what technological errors affect the possible errors in such DPAs.
Author response: Following the reviewer’s comment a paragraph is added to mention the complexity of the work.
Author action: We updated the manuscript by adding the technological points and the operational instructions on page 7 and the second paragraph.
Thank you.
Best Regards,
Kang-Yoon Lee
Reviewer 2 Report
I think this paper is an excellent digital power amplifier with high resolution.
Please make a few corrections.
1. The formula 1 and 2 in the PDF file is broken. Looks like it needs correction.
2. It would be good to attach test results using multi-signal. Please indicate the input power level for the measurement result as well.
3. Are there any measurement results for IIP3 or OIP3?
4. Structurally, in the case of LB PATH, is there a reason to put the attenuator first? Are there any advantages?
Author Response
Concern # 1: The formula 1 and 2 in the PDF file is broken. Looks like it needs correction.
Author response: The reviewer’s comment is concerned and after performing the revisions a corrected PDF file is provided.
Author action: We updated the manuscript following the comments and finally a PDF file is provided correctly.
Concern # 2: It would be good to attach test results using multi-signal. Please indicate the input power level for the measurement result as well.
Author response: The input power of the DPA comes from the transmitter chain, DAC, BBA, and the up-conversion mixer. Therefore, there is no any control or any test signal in the measurement to have an accurate estimation of the input power level for the DPA.
Author action: We updated the manuscript by adding more description to make the measurement conditions more clear, on page 8, the first paragraph, and the 11th line.
Concern # 3: Are there any measurement results for IIP3 or OIP3?
Author response: Because the DPA is followed by an external PA on the test board, the output of the external PA is analyzed. The other issue is due to the full-chain operation of the transmitter which does not give any control over the input power of the DPA to perform any additional measurements.
Concern # 4: Structurally, in the case of LB PATH, is there a reason to put the attenuator first? Are there any advantages?
Author response: In the Low Power Path of both the LB and HB DPAs, the attenuators are placed to provide the attenuation steps before the power cores. Therefore, the attenuators are isolated from the output node of the PA and the balun connection and will not provide any loss or does not influence the operation of the DPA in the other gain steps like amplification gain-steps.
By putting the attenuators after the power cores, the output of the DPA (the balun) are influenced and the quality of the balun degrades. They will introduce an additional loss at the output as well.
Thank you for your comments,
Best Regards,
Kang-Yoon Lee
Round 2
Reviewer 1 Report
The authors made some changes to the article.
I think the work can be published.
Author Response
Dear Reviewer,
Thank you very much.
Best regards,
Kang-Yoon Lee